# Management of Out-of-Hospital Cardiac Arrest during COVID-19: A Tale of Two Cities

**DOI:** 10.3390/jcm11175177

**Published:** 2022-09-01

**Authors:** Shir Lynn Lim, Lekshmi Kumar, Seyed Ehsan Saffari, Nur Shahidah, Rabab Al-Araji, Qin Xiang Ng, Andrew Fu Wah Ho, Shalini Arulanandam, Benjamin Sieu-Hon Leong, Nan Liu, Fahad Javaid Siddiqui, Bryan McNally, Marcus Eng Hock Ong

**Affiliations:** 1Department of Cardiology, National University Heart Centre, Singapore 119228, Singapore; 2Department of Medicine, National University of Singapore, Singapore 119228, Singapore; 3Pre-Hospital and Emergency Research Centre, Health Services and Systems Research, Duke-NUS Medical School, National University of Singapore, Singapore 169857, Singapore; 4Department of Emergency Medicine, Section of Prehospital and Disaster Medicine, Emory University, Atlanta, GA 30322, USA; 5Center for Quantitative Medicine, Duke-NUS Medical School, National University of Singapore, Singapore 169857, Singapore; 6Department of Emergency Medicine, Singapore General Hospital, Singapore 169608, Singapore; 7Rollins School of Public Health, Emory University, Atlanta, GA 30322, USA; 8Emergency Medical Services Department, Singapore Civil Defence Force, Singapore 408827, Singapore; 9Emergency Department, National University Hospital, Singapore 119074, Singapore

**Keywords:** pandemic, cardiac arrest, pre-hospital interventions, survival to hospital discharge

## Abstract

Variations in the impact of the COVID-19 pandemic on out-of-hospital cardiac arrest (OHCA) have been reported. We aimed to, using population-based registries, compare community response, Emergency Medical Services (EMS) interventions and outcomes of adult, EMS-treated, non-traumatic OHCA in Singapore and metropolitan Atlanta, before and during the pandemic. Associations of OHCA characteristics, pre-hospital interventions and pandemic with survival to hospital discharge were analyzed using logistic regression. There were 2084 cases during the pandemic (17 weeks from the first confirmed COVID-19 case) and 1900 in the pre-pandemic period (corresponding weeks in 2019). Compared to Atlanta, OHCAs in Singapore were older, received more bystander interventions (cardiopulmonary resuscitation (CPR): 65.0% vs. 41.4%; automated external defibrillator application: 28.6% vs. 10.1%), yet had lower survival (5.6% vs. 8.1%). Compared to the pre-pandemic period, OHCAs in Singapore and Atlanta occurred more at home (adjusted odds ratio (aOR) 2.05 and 2.03, respectively) and were transported less to hospitals (aOR 0.59 and 0.36, respectively) during the pandemic. Singapore reported more witnessed OHCAs (aOR 1.96) yet less bystander CPR (aOR 0.81) during pandemic, but not Atlanta (*p* < 0.05). The impact of COVID-19 on OHCA outcomes did not differ between cities. Changes in OHCA characteristics and management during the pandemic, and differences between Singapore and Atlanta were likely the result of systemic and sociocultural factors.

## 1. Introduction

The COVID-19 pandemic has had a varying impact on out-of-hospital cardiac arrest (OHCA) globally [1,2]. Regions severely affected by the pandemic reported lower rates of successful pre-hospital resuscitation and increased mortality for OHCA, thought to be related to sicker patients, changes in OHCA characteristics and health-providing behavior of the public, and disruptions in Emergency Medical Services’ (EMS) and hospitals’ systems-of-care [3,4,5]. Despite these observations, some regions less affected by the pandemic reported worse outcomes despite a manageable COVID-19 load [6,7]; while other communities reported minimal change in the epidemiology of OHCAs [8,9].

Regional variations in OHCA epidemiology, poorer outcomes observed amongst communities less affected by the pandemic may be related to the pandemic response, underlying population characteristics and sociocultural factors, as well as the efficiency of pre-hospital care, healthcare access and healthcare delivery during the pandemic. Understanding the influence of these regional characteristics between geographically confined areas could allow for more targeted public health interventions and policy implementation as the pandemic continues. 

The developed island city-state of Singapore and metropolitan Atlanta are fairly similar in population sizes (5.7 million and 4.2 million, respectively) but differ in terms of EMS systems, city architecture and sociocultural factors. These cities had similar COVID-19 infection numbers (29,320 and 29,005 official COVID-19 cases during the study period, translating into incidence rates of 514 per 100,000 population and 697 per 100,000 population, respectively) [10,11], imposed public health restrictions to reduce COVID-19 transmission, and had different COVID-19 case-fatality rates (0.08% and 3.6%, respectively). These similarities and differences presented a unique opportunity for comparison. The purpose of this study was, using population-based registries, to compare the impact of the COVID-19 pandemic on the community response, EMS interventions and outcomes of OHCA before and during the pandemic in Singapore and Atlanta. 

## 2. Materials and Methods

### 2.1. Study Design and Setting

This before–after comparison study included adult (≥18 years old), EMS-treated, non-traumatic OHCA occurring in Singapore between 23 January to 20 May (Figure 1a) and Atlanta between 2 March to 28 June in 2019 and 2020 (Figure 1b). The periods in 2020 reflected the individual cities’ first 17 weeks from the first official COVID-19 case and were chosen to reflect the early response to the COVID-19 pandemic.

Singapore, a multi-ethnic city-state in the Asia-Pacific, operates a single nationwide EMS system through the Singapore Civil Defense Force (SCDF) [12]. Each OHCA case is attended by three Emergency Medical Technicians (EMT); one is EMT-Intermediate (EMT-I) equivalent and two EMT-Basic (EMT-B) equivalent, with one as the ambulance driver. Motorcycle-based EMTs or fire bikers are dispatched ahead of ambulances when available. A series of interventions were introduced over the years to improve the overall pre-hospital response to OHCA—dispatch-assisted cardiopulmonary resuscitation (CPR) in 2012, community first responder scheme in 2014, termination of resuscitation (TOR) in 2019 and a tiered response to OHCA in 2019 [13,14].

Singapore reported its first case of COVID-19 on 23 January 2020 [15]. It raised its Disease Outbreak Response System Condition alert to the second highest level, “orange” on 7 February 2020 and enforced a partial national lockdown from 3 April 2020 to 2 June 2020 in response to an increasing number of infections [16,17].

Metropolitan Atlanta is the most populous metro area in the state of Georgia with a total population of 4.2 million in 8 counties and is served by 13 EMS agencies [18]. These are staffed by a combination of EMT-Intermediate (EMT-I), EMT-Advanced (EMT-A) and paramedics (EMT-P), to provide a multi-tiered response to OHCA. 

Atlanta documented its first case of COVID-19 on 2 March 2020, in Fulton county [19]. A public state health of emergency was declared in Georgia (GA) on 14 March 2020 and lasted beyond the study period [20].

Table 1 details the geography, EMS system and response to COVID-19 pandemic in Singapore and Atlanta.

### 2.2. Data Sources

Data for Singapore were imported from the Pan-Asian Resuscitation Outcomes Study (PAROS) database. PAROS is a prospective, multi-center registry which provides baseline information on OHCA epidemiology, management and outcomes in the Asia-Pacific [24]. Data are extracted from emergency dispatch records, ambulance case notes, and emergency department and in-hospital records. 

The Cardiac Arrest Registry to Enhance Survival (CARES) is a prospective multi-center registry of patients with EMS-treated OHCA in the United States with a catchment area of approximately 167 million residents, established by the United States Centers for Disease Control and Prevention (CDC) and Emory University [25,26]. Data are collected from 3 sources: 911 dispatch centers, EMS agencies, and receiving hospitals. Only data from metropolitan Atlanta were used for this study. 

### 2.3. Data Elements and Definitions

All data definitions for PAROS and CARES are in accordance with Utstein definitions [27]. The total response time (in minutes) referred to the interval between time call received by the dispatch center and the time of patient contact by either the ambulance or rapid responder dispatched via the same dispatch center.

The COVID-19 pandemic period referred to the first 17 weeks from the first confirmed COVID-19 case in each state, which was 23 January 2020 for Singapore and 2 March 2020 for Atlanta. The pre-pandemic period referred to the same 17 weeks of the preceding year. The primary outcome was survival to hospital discharge, defined as discharge from acute hospital care. Secondary outcomes included: (1) transport to acute hospital, (2) survival to hospital admission, defined as admission to hospital intensive care unit after successful resuscitation in the emergency department, and (3) neurological status at time of hospital discharge, based on the Cerebral Performance Category (CPC) scale, where CPC 1 or 2 denoted a positive neurological outcome and CPC 3 or 4 denoted a poor neurological outcome. Inpatient mortality was designated CPC 5. 

### 2.4. Statistical Analysis

Demographics and baseline characteristics of adult EMS-treated, non-traumatic OHCA patients were reported for pandemic and pre-pandemic periods in Singapore and Atlanta as median (first and third quartile (Q1, Q3)) and frequency (percentage) for continuous and categorical variables, respectively. For model building, the variables with multiple categories or levels were re-categorized, and synchronized with the objectives of the study such that their interpretation made clinical and practical sense. Pandemic vs. pre-pandemic was considered as a binary outcome in a logistic regression model, where OHCA characteristics were compared between the pandemic period and pre-pandemic period using multivariable logistic regression analysis in Singapore and Atlanta separately, accounting for potential confounders. Potential confounders, including location type of arrest, witnessed arrest, bystander CPR performed, bystander automated external defibrillator (AED) applied, were chosen based on statistical significance (univariate *p* value < 0.2) and clinical relevance. Statistically significant variables (as potential confounders) were assessed via univariate logistic regression analysis with a less conservative threshold of *p* < 0.2 to allow identification of potential confounders to adjust for in the multivariable model. The same methodology was used to compare the odds of clinical outcomes (survival to hospital discharge, transport to acute hospital, survival to hospital admission and neurological status at discharge) between the two time periods while adjusting for age, gender, location type, witnessed arrest, bystander interventions, first rhythm of arrest, pre-hospital defibrillation and total response time for each city. Odds ratios (OR) and 95% confidence intervals (CI) of observing a characteristic or an outcome between the two periods were calculated. Clinical meaningful interactions were also explored for each city by testing clinically meaningful interaction terms in the multivariable logistic regression models. The impact of the pandemic on OHCA characteristics and outcomes were compared between the two cities by including the interaction term of period and city in the multivariable logistic regression analysis. Model goodness-of-fit was assessed by the Hosmer–Lemeshow test. Significance level was set at *p*-value < 0.05. Statistical analyses were performed using SAS software version 9.4 for Windows (Cary, NC, USA: SAS Institute Inc.).

## 3. Results

### 3.1. Overall Characteristics

The overall study population comprised 3984 EMS-treated OHCA with a median (Q1, Q3) age of 69 (58, 80) years and 2396 (60.1%) males, of which 2084 occurred during the pandemic and 1900 during the pre-pandemic period. The racial distribution is summarized in the Appendix A. 

Baseline characteristics of EMS-treated OHCA in Singapore and Atlanta are summarized in Table 2. The majority of OHCA occurred at home and were of presumed cardiac etiology in both Singapore and Atlanta; almost half of the OHCAs were unwitnessed. Compared to patients in Atlanta, those in Singapore were older (median age of 72 vs. 66), with a higher proportion of males (64.1% vs. 56.2%) and received more bystander interventions (CPR: 65.0% vs. 41.4% and AED application: 28.6% vs. 10.1%). A higher proportion of patients were transported to acute hospitals in Singapore (92.2% vs. 80.9%) but the proportion of patients who survived to hospital discharge in Singapore was less than that reported in Atlanta (5.6% vs. 8.1%). 

### 3.2. Changes in OHCA Epidemiology against the Backdrop of the COVID-19 Pandemic 

The peak of the COVID-19 pandemic in Singapore coincided with increased numbers of OHCA, reduced bystander CPR, reduced transport to acute hospitals and lower survival to hospital discharge rates. The subsequent weeks saw improvements mainly in rates of bystander CPR and transport to acute hospitals, with marginal improvements in survival to hospital admission and discharge (Figure 2a). In Atlanta, changes in OHCA pre-hospital care and outcomes were less congruent with the trajectory of the COVID-19 pandemic. An obvious dip in the rates of transport to acute hospital was seen in mid-April but this was not accompanied by changes in OHCA numbers, or rates of bystander CPR, survival to hospital admission and discharge. The increase in COVID-19 infections in late June was accompanied by increase in OHCA numbers and reductions in rates of bystander CPR, survival to hospital admission and discharge (Figure 2b).

### 3.3. Descriptive Comparison between Pandemic and Pre-Pandemic Periods in Singapore and Atlanta

The pandemic period saw changes in OHCA characteristics, pre-hospital interventions and outcomes, when compared to the pre-pandemic period (Table 3). In Singapore, more OHCAs occurred at home (79.8% vs. 75.2%) and were witnessed (61.1% vs. 45.5%). Fewer received bystander CPR (62.5% vs. 67.7%) and AED application (21.3% vs. 33.1%). The total response and scene times in Singapore were longer during the pandemic (median 12.8 vs. 11.3 min and 24.8 vs. 22.8 min, respectively). Fewer OHCAs were transported to acute hospitals (90.3% vs. 94.2%), survived to hospital admission (13.5% vs. 17.9%) and discharged alive (4.4% vs. 6.9%). Atlanta similarly reported more OHCAs occurring at home during the pandemic (73.9% vs. 66.2%), longer EMS scene time (median 23.0 vs. 20.3 min), lower proportion of OHCA transported to hospitals (75.0% vs. 87.7%), fewer survivals to hospital admission (19.4% vs. 23.6%) and fewer discharged alive (7.3% vs. 9.1%). In contrast to Singapore, the pandemic saw fewer witnessed OHCA in Atlanta (50.7% vs. 53.4%). and little changes in the rates of bystander CPR and AED application.

The changes observed during the pandemic differed according to the location of OHCA. In Singapore, the increased proportion of witnessed OHCA and total response time, as well as lower proportion of transport to acute hospitals reported during the pandemic were largely contributed by OHCA occurring at home (Appendix A). The decrease in bystander CPR during the pandemic was seen in OHCA occurring at home and in public, but more marked for OHCA in public. In Atlanta, the slight decrease in witnessed OHCA during the pandemic was contributed largely by OHCA occurring in public areas; OHCA occurring in public areas reported a decrease in bystander CPR while those occurring at home received more bystander CPR during the pandemic (Appendix A). The decline in transport to acute hospitals was largely in OHCA with no return of spontaneous circulation (ROSC) resulting in field termination in a non-public setting. 

### 3.4. Comparison between Pandemic and Pre-Pandemic Periods in Singapore and Atlanta by Logistic Regression

Some of these differences in OHCA characteristics and pre-hospital care persisted in subsequent analyses with logistic regression (Table 4). Adjusting for clinical, circumstantial and interventional characteristics of an OHCA patient, the odds of being transported to acute hospitals were lower in Singapore (aOR 0.59; 95% CI: 0.41–0.85) and Atlanta (aOR 0.36; 95% CI: 0.26–0.50), and the odds of surviving to hospital admission showed a near-significant decline in Singapore (aOR 0.74; 95% CI: 0.54–1.00) during the pandemic period. The odds of surviving to hospital discharge and reporting a good neurological outcome at discharge were not significantly different (pandemic vs. pre-pandemic) in Singapore (aOR 0.72; 95% CI: 0.43–1.20 and aOR 0.64; 95% CI: 0.37–1.13) and Atlanta (aOR 1.10; 95% CI: 0.71–1.71 and aOR 1.02; 95% CI: 0.61–1.69) during the pandemic.

### 3.5. Comparison of the Impact of COVID-19 Pandemic between Singapore and Atlanta

The impact of the COVID-19 pandemic on OHCA characteristics was significantly different in the two cities (Table 4). In Singapore, the odds of having a witnessed arrest were higher during the pandemic period (aOR 1.96; 95% CI:1.59–2.40) yet it was less likely to receive bystander CPR (aOR 0.81; 95% CI: 0.66–0.99); these were not observed in Atlanta (*p* < 0.001 and *p* = 0.042, respectively). There were no significant differences in the impact of the COVID-19 pandemic on OHCA outcomes between Singapore and Atlanta. 

## 4. Discussion

This East–West collaborative study across similar yet distinct geographical, systems and sociocultural borders saw differences in the impact of the COVID-19 pandemic on community response and EMS systems-of-care for OHCA in two cities, which were less severely affected by the pandemic. While Singapore reported lower bystander CPR and AED application rates, longer total response times and lower transport rates, only the latter was evident in Atlanta. The proportion of survival to hospital discharge was reduced in Singapore, albeit not statistically significant. Atlanta reported no significant difference in survival to hospital discharge. Our study extends the findings of prior studies by providing more granular information on how the COVID-19 pandemic affected pre-hospital management of OHCA and highlights the complex interplay of systems and sociocultural factors in explaining the variations observed. 

Residential OHCA predominated in both cities, and increased during the pandemic. Singapore saw a corresponding increase in the proportion of witnessed arrests but paradoxical decrease in bystander CPR, whereas Atlanta reported similar rates of witnessed residential OHCA during the pandemic, with more receiving bystander CPR. The inter-generational living arrangements prevalent in Singapore could have resulted in more witnessed residential OHCA, and it is plausible that family members witnessing the arrests may not perform CPR due to a combination of knowledge deficits, as well as cultural and psychological barriers [28]. Efforts to improve residential OHCA outcomes must be looked into, and these include educating and empowering family members to perform CPR, and improving EMS response times.

The predominance of residential OHCA also highlighted the influence of city architecture on care delivery by EMS personnel. Singapore is heavily urbanized where 90% of its population resides in high-rise apartments. OHCAs occurring in high-rise buildings are a challenge to contemporary EMS [29]; this is reflected in the doubling of time of scene arrival to patient access in Singapore compared with Atlanta, and greater delays during the pandemic in Singapore. Protocols to override elevator systems and staying on-scene for the delivery of optimal basic and advanced life support to achieve ROSC may help improve OHCA outcomes. 

The COVID-19 pandemic necessitated changes in EMS workflows [23,30], which may have conserved resources and protected EMS personnel from unnecessary exposure, but potentially reduced the likelihood of successful resuscitation. Firstly, the need to don PPE prior to attending to any EMS calls (Singapore) or high-risk calls (Atlanta) likely contributed to increased EMTs’ fatigue level and lengthened EMS response times during the pandemic. Limiting the number of EMS personnel dispatched to site reduced the efficacy and efficiency of pre-hospital resuscitation (i.e., no high-performance CPR in Singapore). Finally, protocols recommending transporting only patients with ROSC may have contributed to lower transport rates seen during the pandemic, particularly in Atlanta. Despite these, we were reassured by the non-significant changes in survival to hospital discharge and good neurological outcomes on discharge. 

The disruptions in health provision at the pre-hospital level observed despite the relatively low COVID-19 case-fatality rates for Singapore and Atlanta during the study period [10,11] called into question the suspension of community first-responder schemes and changes to EMS protocols early in the pandemic. It is plausible that reduced bystander CPR rates observed in Singapore during the pandemic were partly contributed to by the suspension of community-first responder schemes. Although both cities reported non-significant changes in OHCA outcomes as a result of the COVID-19 pandemic, the longer-term impact of these changes on successful pre-hospital resuscitation and eventual OHCA outcomes is unknown and deserves further study. Variability and change over time are also a part of any pandemic; hence, governments, related agencies and key stakeholders must continually assess the local situation and adapt their response.

The strengths of our study include the population-based design of both databases with data collection based on Utstein definitions for reporting cardiac arrest. Both databases used have in-built quality control measures, therefore ensuring data quality and integrity. Nonetheless, our study should be interpreted in the context of the following limitations. Our before–after study design limited our ability to control for secular trends. As we included only EMS-treated adult OHCA in our study, we could not comment on the impact of COVID-19 on overall OHCA incidence or the proportion who received treatment. As both registries collected mainly essential pre-hospital data variables and hospital outcomes, we lacked information on aetiology of arrest, socioeconomic factors and hospital-based management. EMS timings were not available for ~40% of the Atlanta cohort as these were optional data, limiting comparison. Finally, our findings may not be generalizable to regions with different COVID-19 trajectories, particularly low- and middle-income countries severely affected by COVID-19. 

## 5. Conclusions

Changes in OHCA characteristics and pre-hospital interventions in Singapore and Atlanta during the COVID-19 pandemic were likely collateral consequences, with differences between cities partly reflecting differences in systems-of-care and other sociocultural factors. These highlight opportunities for public education and mutual exchange of knowledge from different systems. Further studies into lower bystander intervention and EMS transport rates during the pandemic will help build a more resilient OHCA EMS response capable of weathering current and future pandemics.

## Figures and Tables

**Figure 1 jcm-11-05177-f001:**
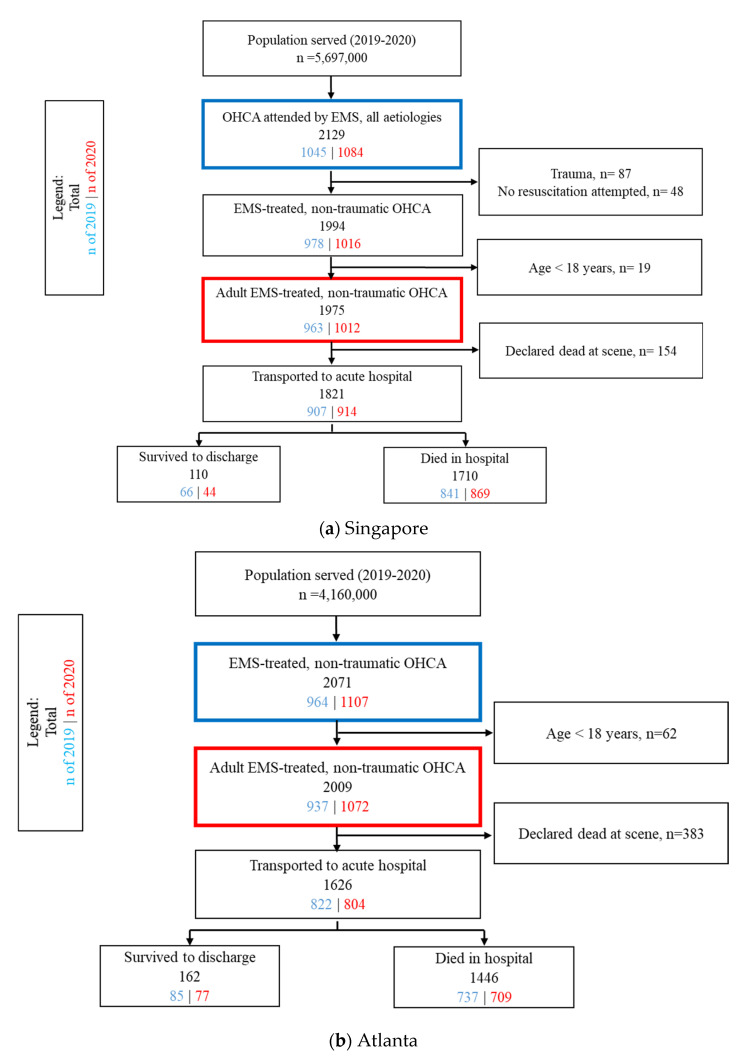
Flowchart of patient selection in Singapore and Atlanta. Patient selection during the pandemic (17 weeks from date of first confirmed COVID-19 case in 2020) and pre-pandemic (corresponding dates in 2019) periods. For (**a**) Singapore, the date of the first confirmed case was 23 January 2020, and (**b**) Atlanta, the date of the first confirmed case was 2 March 2020. The blue box indicates OHCA patients captured by the respective registries; the red box indicates the final study population. Outcome (survival) data were not available for 1 patient in Singapore and 18 patients in Atlanta. Abbreviations: OHCA, out-of-hospital cardiac arrest; EMS, Emergency Medical Services; ROSC, return of spontaneous circulation; COVID-19, coronavirus disease 2019.

**Figure 2 jcm-11-05177-f002:**
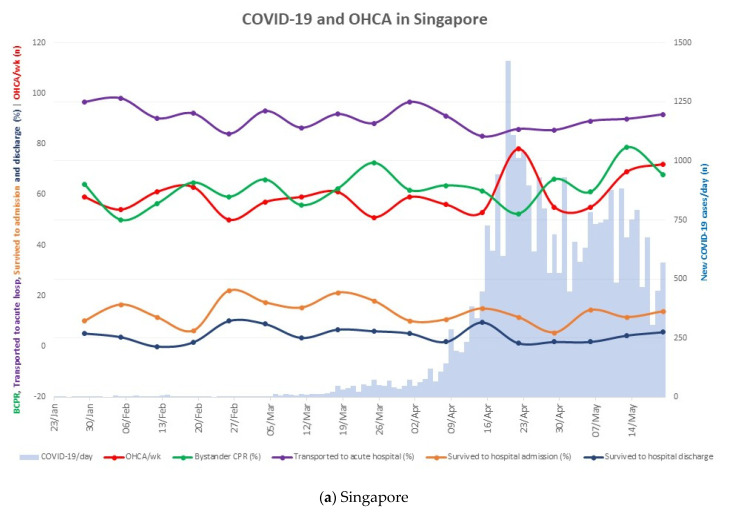
COVID-19 and OHCA in Singapore and Atlanta. Pre-hospital interventions for and outcome of OHCA against a backdrop of COVID-19 pandemic in (**a**) Singapore, and (**b**) Atlanta. X-axis depicts the first 17 weeks (119 days) of the pandemic, starting on 23 January 2020 in Singapore and 2 March 2020 in Atlanta. Y-axis on the left depicts the weekly average of OHCA (n), bystander CPR (%), OHCA transported to acute hospitals (%), survival to hospital admission (%) and survival to hospital discharge (%). Y-axis on the right depicts the daily number of new COVID-19 cases. Abbreviations: COVID-19, coronavirus disease 2019; OHCA, out-of-hospital cardiac arrest; BCPR, bystander cardiopulmonary resuscitation.

**Table 1 jcm-11-05177-t001:** Differences between Singapore and Atlanta.

	Singapore	Atlanta
**Geography**		
Land size	728.3 km^2^	7587.6 km^2^ 8 counties (Cobb, Clayton, DeKalb, Douglas, Fulton, Gwinnett, Newton and Rockdale)
Population (2019 estimates) [18,21]	5,704,000	4,160,864
Population density	7832 persons per km^2^	548 persons per km^2^
**EMS systems**		
Number of agencies	One national EMS agency, the Singapore Civil Defense Force Fire-based system activated by a centralised 995 dispatch system	13 EMS agencies serving these 8 counties A combination of fire-based, hospital-based, third party and volunteer systems A centralised 911 PSAP/ECC connects the call to the agency serving the area
Response to OHCA	Community first responders activated by mobile applications Multi-tier response to OHCA commenced April 2019 - First responders: EMT-B on firebikes - Ambulance staffed by 2 EMT-B equivalent and 1 EMT-I equivalent - Additional fire medical vehicles for enhanced medical support, including high performance CPR Transport to the nearest restructured hospital Protocols for withholding and terminating resuscitation, with the latter commencing in January 2019	EMS providers are EMT-I, EMT-A and paramedics Multi-tier response to OHCA - The first responder, usually fire-based or volunteer staffed by EMT-I/EMT-A or paramedic to initiate resuscitation - The transporting agency simultaneously dispatches an ALS capable ambulance that has a paramedic as the highest-level provider. Encouraged to resuscitate in place and transport once ROSC obtained unless witnessed arrest, traumatic arrest, refractory VF or public setting. OHCA patients are transported to appropriately resourced Emergency Cardiac Centres (designated by Levels ie Levels I, II and III).
Training/Skills	EMT-B equivalents need to undergo 5 weeks of training. They are BLS-certified and able to carry out defibrillation. EMT-I equivalents require 15 months of training, and are able to administer IV and IO drugs, as well as insert laryngeal mask airway.	EMT-I/A undergo 20 weeks; 303 contact hours of training. They are BLS-certified and able to use an AED, insert supraglottic airway, IV/IO and administer fluids and dextrose. Paramedics undergo 16 months; minimum 1084 contact hours of training and are able to provide ALS level of care including manual defibrillation, intubation, IV/IO and administer ALS medication including epinephrine/amiodarone and atropine.
**COVID-19 Epidemiology ***		
Incidence	29,320 cumulative new cases Incidence rate of 514 per 100,000 population [10]	29,005 cumulative new cases Incidence rate of 697 per 100,000 population [11]
Mortality	22 deaths Case-fatality rate of 0.08%	1034 deaths Case-fatality rate of 3.6%
**Response to COVID-19**		
Public	Disease Outbreak Response System Condition (DORSCON) raised to Orange on 7 February 2020 Additional public health measures and travel advisories imposed on 6 March in response to increasing community transmissions Partial national lockdown from 3 April to 2 June 2020 - Hospitals halted non-critical services - School closures - Safe distancing regulations - Closures of beaches and playgrounds Mandatory mask wearing imposed on all >2 years of age from 14 April 2020, which is still in place	Public health state of emergency declared in Georgia on 14 March 2020 (last beyond study period) - Social distancing recommended - Increased COVID-19 testing capabilities - Building of isolation zones Closure of public elementary, secondary and post-secondary schools in Georgia from 18 to 31 March 2020. This was subsequently extended through the end of 2019–2020 school year. Additional measures: isolation, quarantine and shelter regulations, increased social distancing measures Gradual re-starting of the economy from 24 April 2021
EMS	Non-emergent, COVID-19 suspect cases were managed by a separate dedicated fleet of ambulances managed by a separate call center (operated by centralised “993” dispatch system) Single-tier response to OHCA from 7 February 2020 onwards, where fast response bikes and fire appliances stopped being deployed All ambulance personnel operate in full PPE for every emergency case attended All ambulance personnel to don PPE prior	Modified caller queries about SARS-CoV-2 infection - 911 PSAP/ECCs should question callers and determine whether the call concerns a person who might have COVID-19 - Information about a patient who might have COVID-19 should be communicated immediately to EMS personnel before arrival on scene in order to limit the number of EMS personnel exposed to the patient and to allow use of appropriate PPE Universal source control measures - Patients and family members should be wearing their own cloth face covering prior to the arrival of EMS personnel and throughout the duration of the encounter, including during transport - EMS personnel should wear a face mask at all times while they are in service ** Universal use of PPE in areas with moderate to substantial community transmission, optional in areas with low community transmission [22] Encourage physical distancing - Limiting the number of EMS personnel (to essential personnel) in the patient compartment during transport - Limiting those riding in the ambulance while patient is transported to those essential for the patient’s care Guidance on management of those with suspected or confirmed COVID-19, including PPE, aerosol-generating procedures, advanced life support, transport to a healthcare facility and cleaning of vehicle following transport [23]

* COVID-19 epidemiology from 23 January 2020 to 20 May 2020 for Singapore, and 2 March 2020 to 28 June 2020 for Atlanta. ** Some EMS agencies had employees wear full PPE (N95 masks and face shield or goggles for every patient contact regardless of suspicion of COVID-19 status). Abbreviations: EMS, Emergency Medical Services; PSAP, Public Safety Answering Points; ECC, Emergency Communication Centres; EMT-B, Emergency Medical Technician-Basic; EMT-I, Emergency Medical Technician-Intermediate; EMT-A, Emergency Medical Technician-Advanced; OHCA, out-of-hospital cardiac arrest; ALS, advanced life support; BLS, basic life support; AED, automated external defibrillator; IV, intravenous; IO, intraosseous; COVID-19, Coronavirus Disease 2019; SARS-CoV-2, Severe Acute Respiratory Syndrome Coronavirus 2; PPE, personal protective equipment.

**Table 2 jcm-11-05177-t002:** Baseline characteristics of EMS-treated OHCA patients.

	Singapore N = 1975	Atlanta N = 2009	*p*-Value ***
**Demographics**			
Age in years, median [Q1, Q3]	72.0 [61.0, 83.0]	66.0 [54.0, 76.0]	<0.001
Male gender, n (%)	1266 (64.1%)	1130 (56.2%)	<0.001
**Event information, n (%)**			
Arrest location			<0.001
- private residence	1532 (77.6%)	1412 (70.3%)	
- healthcare facility	183 (9.3%)	369 (18.4%)	
- public area	260 (13.2%)	228 (11.3%)	
Presumed cardiac aetiology	1781 (90.2%)	1742 (86.7%)	<0.001
Initial shockable rhythm	316 (16.0%)	333 (16.6%)	0.654
Witnessed arrest			
- Unwitnessed	919 (46.5%)	966 (48.1%)	
- Bystander witnessed	858 (43.4%)	758 (37.7%)	<0.001
- EMS witnessed	198 (10.0%)	285 (14.2%)	
**Pre-hospital resuscitation, n(%)**			
Bystander CPR	1049 (65.0%)	574 (41.4%)	<0.001
Bystander AED application	66 (28.6%)	20 (10.1%)	<0.001
Pre-hospital defibrillation	462 (23.4%)	535 (26.6%)	0.020
**EMS response times in min, median [Q1, Q3) ***			
EMS response time	8.28 [6.76, 10.2]	9.00 [6.43, 12.0]	<0.001
Total response time	12.0 [10.0, 14.5]	11.0 [8.38, 14.2)	<0.001
- Call received to dispatch	2.07 [1.53, 2.78]	0.633 [0.133, 1.39]	<0.001
- Dispatch to scene arrival	6.07 [4.70, 7.92]	7.74 [5.12, 10.4]	<0.001
- Scene arrival to patient’s side	3.35 [2.07, 4.87]	1.45 [0.917, 2.66]	<0.001
Time at scene	23.9 [20.5, 27.5]	21.8 [16.0, 29.0]	0.939
**Patient outcomes, n (%) ****			
Transported	1821 (92.2%)	1626 (80.9%)	<0.001
Survived to hospital admission	308 (15.6%)	425 (21.3%)	<0.001
Survived to hospital discharge	110 (5.6%)	162 (8.1%)	0.002
Discharged with good neurological outcome	93 (4.7%)	114 (5.7%)	0.174

Numbers are *n* (%) for categorical variables and median [Q1, Q3] for continuous variables. Bystander CPR is defined as CPR performed by a layperson (excludes EMS-witnessed OHCA and OHCA occurring in healthcare facilities). Bystander AED application is defined as AED application by a layperson (excludes EMS-witnessed and non-public area OHCA). * Data from Atlanta are not available for: 822 (40.9%) call received to dispatch; 467 (23.2%) dispatch to scene arrival; 830 (41.3%) scene arrival to patient’s side; 816 (40.6%) EMS response time; 820 (40.8%) total response time; 738 (36.7%) scene time. * Data from Singapore are not available for: 154 (7.8%) scene time. ** Data for survival to hospital discharge are not available for 1 patient from Singapore and 18 patients from Atlanta. *** Statistically significant at *p* < 0.05. Abbreviations: EMS, emergency medical services; OHCA, out-of-hospital cardiac arrest; Q1, first quartile; Q3, third quartile; CPR, cardiopulmonary resuscitation; AED, automated external defibrillator.

**Table 3 jcm-11-05177-t003:** Characteristics of EMS-treated OHCA and outcomes, by city and period.

	Singapore		Atlanta	
	Pandemic N = 1012	Pre-Pandemic N = 963	Pandemic N = 1072	Pre-Pandemic N = 937
**Demographics**				
Age in years, median [Q1, Q3]	73.0 [61.0, 84.0]	72.0 [60.0, 83.0]	66.0 [54.0, 76.0]	66.0 [54.0, 77.0]
Male gender, n (%)	654 (64.6%)	612 (63.6%)	581 (54.2%)	549 (58.6%)
**Event information, n (%)**				
Arrest location				
- home residence	808 (79.8%)	724 (75.2%)	792 (73.9%)	620 (66.2%)
- healthcare facility	99 (9.8%)	84 (8.7%)	190 (17.7%)	179 (19.1%)
- public area	105 (10.4%)	155 (16.1%)	90 (8.4%)	138 (14.7%)
Presumed cardiac aetiology	928 (91.7%)	853 (88.6%)	913 (85.2%)	829 (88.5%)
Initial shockable rhythm	158 (15.6%)	158 (16.4%)	163 (15.2%)	170 (18.1%)
Witnessed arrest				
- Unwitnessed	394 (38.9%)	525 (54.5%)	529 (49.3%)	437 (46.6%)
- Bystander witnessed	510 (50.4%)	348 (36.1%)	392 (36.6%)	366 (39.1%)
- EMS witnessed	108 (10.7%)	90 (9.4%)	151 (14.1%)	134 (14.3%)
Pre-hospital resuscitation, n (%)				
Bystander CPR	511 (62.5%)	538 (67.7%)	309 (41.7%)	265 (41.1%)
Bystander AED application	19 (21.3%)	47 (33.1%)	7 (9.21%)	13 (10.6%)
Pre-hospital defibrillation	227 (22.4%)	235 (24.4%)	275 (25.7%)	260 (27.7%)
**EMS response times in min, median [Q1, Q3] ***				
EMS response times	8.6 [6.9, 10.5]	8.0 [6.5, 9.8]	9.4 [6.6, 12.3]	9.0 [6.3, 11.9]
Total response time	12.8 [10.8, 15.1]	11.3 [9.34, 13.4]	11.4 [8.8, 14.9]	10.9 [8.0, 13.8]
- Call received to dispatch	2.0 [1.5, 2.8]	2.1 [1.6, 2.8]	0.6 [0.1, 1.1]	0.7 [0.1, 1.6]
- Dispatch to scene arrival	6.3 [4.9, 8.2]	5.9 [4.5, 7.6]	8.0 [5.3, 10.9]	7.1 [5.0, 10.0]
- Scene arrival to patient’s side	3.9 [2.7, 5.5]	2.8 [1.6, 4.0]	1.6 [1.0, 3.0]	1.3 [0.8, 2.2]
Time at scene	24.8 [21.3, 28.5]	22.8 [19.6, 26.4]	23.0 [17.5, 31.0]	20.3 [15.2, 27.3]
**Patient outcomes, n (%) ****				
Transported	914 (90.3%)	907 (94.2%)	804 (75.0%)	822 (87.7%)
Survived to hospital admission	136 (13.5%)	172 (17.9%)	204 (19.4%)	221 (23.6%)
Survived to hospital discharge	44 (4.4%)	66 (6.9%)	77 (7.3%)	85 (9.1%)
Discharged with good neurological outcome	36 (3.6%)	57 (5.9%)	54 (5.1%)	60 (6.4%)

Numbers are *n* (%) for categorical variables and median (Q1, Q3) for continuous variables. Bystander CPR is defined as CPR performed by a layperson (excludes EMS-witnessed OHCA and OHCA occurring in healthcare facilities). Bystander AED application is defined as AED application by a layperson (excludes EMS-witnessed and non-public area OHCA). The EMS response time (in minutes) refers to the interval between time call received by the dispatch center and the time of ambulance arrival at scene. The total response time (in minutes) refers to the interval between time call received by the dispatch center and the time of patient contact by either the ambulance or rapid responder dispatched via the same dispatch center. * Data from Singapore are not available for: 154 (7.8%) scene time. Data from Atlanta are not available for: 822 (40.9%) call received to dispatch; 467 (23.2%) dispatch to scene arrival; 830 (41.3%) scene arrival to patient’s side; 816 (40.6%) EMS response time; 820 (40.8%) total response time; 738 (36.7%) scene time. ** Data for survival to hospital discharge are not available for 1 patient from Singapore and 18 patients from Atlanta. Abbreviations: EMS, emergency medical services; OHCA, out-of-hospital cardiac arrest; CPR, cardiopulmonary resuscitation; AED, automated external defibrillator.

**Table 4 jcm-11-05177-t004:** Multivariable logistic regression of OHCA event characteristics and outcome between pandemic and pre-pandemic periods.

Variable	Event vs. Reference Level	Pandemic vs. Pre-Pandemic				
		Singapore		Atlanta		Singapore vs. Atlanta *
		Adjusted OR (95% CI)	*p* Value	Adjusted OR (95% CI)	*p* Value	*p* Value
**OHCA characteristics ^1^**						
Location type	Home vs. Non-home	2.05 (1.50, 2.80)	<0.001	2.03 (1.47, 2.81)	<0.001	NS
Witnessed arrest	Yes vs. No	1.96 (1.59, 2.40)	<0.001	0.96 (0.77, 1.19)	0.683	<0.001
Bystander CPR	Yes vs. No	0.81 (0.66, 0.99)	0.049	1.07 (0.86, 1.34)	0.536	0.042
**Clinical Outcomes ^2^**						
Transport to acute hospital	Yes vs. No	0.59 (0.41,0.85)	0.005	0.36 (0.26,0.50)	<0.001	0.096
Survived to admission	Yes vs. No	0.74 (0.54, 1.00)	0.053	0.83 (0.63, 1.01)	0.186	NS
Survived to discharge	Yes vs. No	0.72 (0.43, 1.20)	0.208	1.10 (0.71, 1.71)	0.660	NS
Discharged with good neurological outcome	Yes vs. No	0.64 (0.37, 1.13)	0.127	1.02 (0.61, 1.69)	0.948	NS

^1^ Multivariable logistic regression of OHCA characteristics, accounting for age (continuous), gender, first rhythm of arrest, location type, witnessed arrest and bystander CPR. Outcome is taken as the year, with reference year being the pre-pandemic. ^2^ Multivariable logistic regression of outcome, accounting for age (continuous), gender, location type, witnessed arrest, bystander CPR, first rhythm of arrest, pre-hospital defibrillation. Outcome is taken as the year, with reference year being the pre-pandemic. * The impact of pandemic on OHCA characteristics and outcomes were compared between the two cities by including the interaction term of Period × City in the multivariable logistic regression analysis. Abbreviations: OHCA, out-of-hospital cardiac arrest; OR, odds ratio; CI, confidence intervals; CPR, cardiopulmonary resuscitation; NS, not significant.

## Data Availability

The data supporting the findings of this study are available from the corresponding author upon reasonable request, subject to approval by PAROS and CARES.

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
