# Peer review of "Management of Out-of-Hospital Cardiac Arrest during COVID-19: A Tale of Two Cities"

_jcm, 2022, doi:10.3390/jcm11175177_

Round 1

Reviewer 1 Report

Using population-based registries, the Authors analyzed the influence of EMS systems, city architecture and sociocultural factors on community response, EMS interventions and OHCA outcomes in Atlanta and Singapore during the COVID-19 pandemic.

Apart from the abstract, which deserves to be improved, the paper is generally well written and interesting. The differences in prehospital systems are well described. The references are adequate. There are a few typos, which should be addressed by virtue of the copyediting process.

Major comments

1. Abstract: summarizing a full article in a 200-word unstructured abstract is very difficult. It is however also most important since many readers will decide whether they will read the full manuscript according to the contents of the abstract. While the current version of the abstract provides interesting information, I believe that it would be markedly enhanced by using a semi-structured format. In other words, the first sentence should explain the background, and the second should state the objective.

2. Lines 178-187: Rather than providing a written interpretation of the results included in Supplemental Table 1, I believe that inserting the Table itself in the Results section would be appropriate.

Minor comments

3. The same number of decimals should be used when reporting similar figures (lines 54-55: "The developed island city-state of Singapore and metropolitan Atlanta are fairly sim-54 ilar in population sizes (5.7 million and 4.16 million respectively)")

4. Lines 176-177: "EMS-treated OHCA in Singapore and Atlanta were similar in terms of OHCA characteristics" - What is meant by "OHCA characteristics"?

Reviewer 2 Report

The article by Shir Lynn Lim et al. Entitled “Management of out-of-hospital cardiac arrest during COVID-19: a tale of two cities” aimed to compare the impact of COVID-19 pandemic on the community response, EMS interventions and outcomes of OHCA before and during the pandemic in Singapore and Atlanta.

The manuscript is well written and the results are interesting. The abstract summarizes the main points. The aim of the study is stated clearly and the introduction gives enough background. Materials and methods are used appropriate and are adequately described. The results are presented clear and concise. Discussion and conclusions are consistent with the aim, methodology and results. Main finding are summarized and put into the context. Strengths and limitations are sufficiently discussed.

However, there are some minor edits and clarifications needed. I have listed some points for improvement below.

Lines 207-208 (the caption of the Figure 2): “Y-axis on the left depicts the daily number of new COVID-19 cases. Y-axis on the right depicts the weekly average of OHCA “. Check and correct, please.

Line 228 (Table 2) I suppose, in the first row the names of both cities should be in bold. The same in the second row.

Line 270 (Table 3) “OHCA characteristics” should be probably moved under the horizontal line.

Reviewer 3 Report

Overview and general recommendations:

This original article describes the circumstances and outcome of out-of-hospital cardiac arrest during COVID-19 pandemic in Singapore and Atlanta, with performing a population-based study. It also makes a comparison between the two geographical places regarding OHCA during the pandemic. 

Overall, the study addresses a very interesting and actual topic. The article is clear and well-written, however, there are some minor changes that should be applied before publication. Please, find below my detailed comments. 

Introduction section: 

The last paragraph of the Introduction part is very important. However, it needs some more additional data regarding the COVID-19 situation in Singapore and Atlanta. Please describe in more details, how severe the pandemic in these cities was. 

Methods section:

Please explain briefly, why 17 weeks were chosen as study period?

Please describe in statistical analysis, why logistic regression was used. In statistical analysis part you write (lines 153-156): “Potential cofounders,… were chosen based on statistical significance (univariate p value <0.2) and clinical relevance.” Which type of univariate analysis did you use? Why 0.2 p value was here chosen? In Supplementary material you state that the level of significance was set at p<0.05 at the comparison of baseline characteristics. 
